# Cancer Metabolic Subtypes and Their Association with Molecular and Clinical Features

**DOI:** 10.3390/cancers14092145

**Published:** 2022-04-25

**Authors:** Enrico Moiso, Paolo Provero

**Affiliations:** 1Department of Molecular Biotechnology and Health Sciences, University of Turin, 10126 Turin, Italy; 2Broad Institute of MIT and Harvard, Cambridge, MA 02142, USA; 3Department of Neurosciences “Rita Levi Montalcini”, University of Turin, 10126 Turin, Italy; 4Center for Omics Sciences, San Raffaele Scientific Institute IRCSS, 20132 Milan, Italy

**Keywords:** cancer metabolism, genomics, computational biology

## Abstract

**Simple Summary:**

The metabolic alterations characteristic of cancer cells play a significant role in tumors’ natural history and response to therapy. Recent technological advances have allowed the production of unprecedented amounts of data on many types of cancers. We exploited the most comprehensive collection of such data, The Cancer Genome Atlas (TCGA), to systematically investigate the associations between metabolic alterations and other tumor features. We used sets of genes known to be associated with specific metabolic pathways to classify patients into “metabolic subtypes”. Then, we systematically looked for associations between the metabolic subtypes and other tumor features, including histological classification, patient survival, and genome alterations. Our results, while correlative in nature, can provide a guide to the formulation of specific mechanistic hypotheses to be tested experimentally so as to improve our understanding of the biology of cancer and our ability to tailor therapeutic interventions to the specific features of each patient.

**Abstract:**

The alterations of metabolic pathways in cancer have been investigated for many years, beginning long before the discovery of the role of oncogenes and tumor suppressors, and the last few years have witnessed renewed interest in this topic. Large-scale molecular and clinical data on tens of thousands of samples allow us to tackle the problem from a general point of view. Here, we show that transcriptomic profiles of tumors can be exploited to define metabolic cancer subtypes, which can be systematically investigated for associations with other molecular and clinical data. We find thousands of significant associations between metabolic subtypes and molecular features such as somatic mutations, structural variants, epigenetic modifications, protein abundance and activation, and with clinical/phenotypic data, including survival probability, tumor grade, and histological types, which we make available to the community in a dedicated web resource. Our work provides a methodological framework and a rich database of statistical associations, which will contribute to the understanding of the role of metabolic alterations in cancer and to the development of precision therapeutic strategies.

## 1. Introduction

The last few years have witnessed a renewed interest in the metabolism of cancer, with the reprogramming of energy metabolism by cancer cells having been recently added, together with immune evasion, to the fundamental oncogenic processes collected under the heading of “Hallmarks of Cancer” [1].

Today, next-generation sequencing and other high-throughput techniques provide us with the unprecedented opportunity to take the study of the role of metabolic alterations in cancer progression to a new level, complementing the in-depth study of specific instances of metabolic changes with unbiased assessments of cancer metabolism and its relationship with the other hallmarks. In particular The Cancer Genome Atlas (TCGA) provides us with a systematic repository of genomic, epigenomic, transcriptomic, and clinical data on tens of thousands of samples of many tumor types. For example, a recent study [2] demonstrated the close relationship between patterns of copy-number alterations and metabolic phenotypes. Another study [3] showed the clinical relevance of metabolic pathways, as reflected by gene expression profiles of metabolic genes, to patient survival.

In this work, we undertake a systematic analysis of the associations between metabolic profiles of human tumors and their molecular and clinical features. Our aim is to underline the role of metabolism in precision cancer medicine by showing that metabolic profiling can classify tumors into classes that are significantly different in both their molecular-level features (somatic mutations, structural genomic variants, epigenetic modifications) and in their clinical aspects (survival probability, tumor grade, histological type). While intrinsically correlative in nature, this analysis can serve as a guide to mechanistic studies, linking metabolism to other molecular features, and to the exploration of precision therapeutic approaches informed by tumor metabolism.

While, ideally, such a study should be based on direct assays of metabolite abundance in primary tumors, these are not yet available on a large scale. Therefore, we use transcriptomic profiles of tens of thousands of samples of many tumor types, as a proxy for metabolomic assays, to classify them into metabolic subtypes. We then systematically correlate such subtypes with molecular and clinical data using appropriate statistical tests, thus generating a database of thousands of associations.

## 2. Results

### 2.1. Transcriptome-Based Classification of Tumors into Metabolic Subtypes

For each tumor type with transcriptomic data Available online: the TCGA, and for each metabolic pathway defined as a set of genes by annotation databases such as the Kyoto Encyclopedia of Genes and Genomes (KEGG) [4,5], Reactome [6], and the Gene Ontology [7,8], we divided the patients into metabolic subtypes by clustering the samples using the expression profiles of the genes belonging to the pathway. Appendix A shows the KEGG pathways used in the analysis and the genes assigned to each pathway as a bipartite network.

Compared with differential expression analyses, e.g., [3], clustering by metabolic pathways takes into account the fact that the genes associated with a pathway are often involved in either anabolic or catabolic processes. Therefore the activation of the pathway in a subset of patients is typically reflected by the transcriptional activation of the former and repression of the latter. For example, in Figure 1, we show the clustering of low-grade glioma samples based on the “arachidonic acid metabolism” KEGG pathway: each cluster is characterized by both up- and downregulated genes, a structure that would not be captured by simple differential expression of the pathway genes as a whole.

To perform the clustering we used partitioning around medoids (PAM) [9], a more robust method compared to k-means [10] with respect to the presence of outliers, which has been shown to be among the best-performing methods for the classification of cancer samples based on bulk RNA-seq data [11]. A total of 345 metabolic gene sets were used, as described in the Methods, on 38 tumor classes, for a total of 13,110 gene sets/tumor pairs. We used a Duda–Hart test [12] to select the pairs characterized by cluster separation (12,074 pairs, 92.1%). For these pairs, cluster silhouette analysis allowed an unsupervised choice of the number of clusters. In the majority of cases (9157, 75.8%), the samples were subdivided into k=2 clusters, while in other cases we obtained up to 10 clusters. These clusters will be referred to as “metabolic subtypes”. The tumor types analyzed with the number of samples for which expression data were available are shown in Appendix A.

It is worth asking whether different metabolic gene sets cluster the patients of the same tumor in significantly different ways. To answer this question, we computed the normalized mutual information between the cluster assignments obtained in each tumor with each metabolic gene set. The results, for the KEGG pathways analyzed are shown in Figure 2, and suggest that, in most cases, different metabolic gene sets cluster tumor patients in clearly distinct ways.

### 2.2. Associations between Metabolic Subtypes and Molecular and Clinical Features

We then proceeded to systematically explore the statistical associations between metabolic subtypes and several molecular and clinical features of the corresponding tumors, classified into 9 classes of variables (see Appendix A): 6 classes of molecular variables (copy number alterations, global DNA methylation, microRNA expression, specific point mutations, gene-level mutations, and protein expression); and 3 classes of phenotypic ones (clinical and histological parameters appropriate for each tumor type, overall survival, and recurrence-free survival). Statistical tests appropriate to the nature of the variables describing such features were chosen as described in the Methods. Bonferroni correction was applied separately to each variable class but to all tumor types together. Thus, for example, the Bonferroni correction for the association with miRNA expression takes into account all tests of association between all miRNAs and all metabolic gene sets across all tumor types.

A total of $∼4\times 10^{8}$ statistical tests were performed, resulting in 878,655 signifcant associations after Bonferroni correction (0.21% of the tests performed). An overall Bonferroni correction (taking into account together all the tests made for all variable types) would have given 663,100 significant associations. In the rest of this section, to limit redundancy, we focus on the results obtained with the KEGG annotation database (a total of 99,462 significant associations). Figure 3A shows the distribution of these associations among tumor types. Note that the number of significant associations for a given tumor type depends on the number of available samples: indeed, the Spearman correlation coefficient between number of associations and number of samples is 0.755. In Figure 3B, we show the number of associations by KEGG metabolic pathway, suggesting a particular relevance of the metabolism of amino acids and fatty acids in classifying tumors into classes characterized by different molecular and/or clinical characteristics.

### 2.3. Recurrent Associations

Of particular interest are recurrent associations, i.e., those that are statistically significant in more than one tumor type (not considering the tumor types built from the union of smaller types, such as LUNG, KIPAN, etc.). We found a total of 14,899 such recurrent associations. Appendix A shows the most recurrent associations for each class of clinical and molecular variables. Notably, the only variable classes with no recurrent associations are point mutations: all 45 associations with specific mutations and 102 associations with gene-level mutations are specific of a single tumor type.

The most recurrent association was found between the expression of microRNA hsa-miR-222 and the metabolic subtypes based on the “inositol phosphate metabolism” gene set defined by KEGG, which recurred in 12 tumor types. Figure 4 shows the expression of this microRNA in the clusters of breast cancer and lung adenocarcinoma patients obtained with such a gene set.

### 2.4. Specific Examples

In the following, we show a few examples of associations, some confirming known results and some suggesting new lines of investigation, especially in less widely studied tumor types.

#### 2.4.1. TP53 Mutations and Glucose Metabolism

Mutant TP53 has been shown to promote the Warburg effect, possibly the most notorious metabolic hallmark of cancer, in breast and lung cancer cell lines [13]. Indeed, clustering breast and lung tumors using the genes associated with glucose metabolism according to REACTOME shows, in both cancer types, the appearance of a cluster characterized by overexpression of several Warburg effect signature genes (such as GAPDH, PGK1, and PKM2) and by an enrichment in TP53 mutations (Figure 5).

#### 2.4.2. Pyruvate Metabolism in Thyroid Cancer

The clustering of thyroid cancer samples according to the genes associated with KEGG to pyruvate metabolism is shown in Figure 6A. These two clusters show many significant differences in both clinical and molecular parameters. In particular (Figure 6B,C), the prevalence of BRAF and NRAS-mutated samples was strikingly different in the two clusters, with all NRAS-mutated samples found in Cluster 2, and the large majority of BRAF-mutated samples found in Cluster 1. The two clusters also strongly differed in the respective prevalence of histological types.

#### 2.4.3. Fatty Acid Metabolism in Thymoma

Clustering of thymoma samples using the genes associated with fatty acid metabolism divides the samples into two clusters (Figure 7A) largely overlapping the known histological types (Figure 7B) and differing in the prevalence of a recurrent GTF2I mutation [14].

### 2.5. Analysis of Cancer Cell Lines

We performed the same analysis on the cancer cell lines included in the cancer cell line encyclopedia (CCLE) [15]. These cell lines are classified in terms of TCGA tumor types (22 types are represented by a total of 723 CCLE cell lines). Within each tumor type, cell lines were clustered based on their transcriptomic data, using the expression of the genes in the same metabolic gene sets used to cluster TCGA samples. The clusters were then correlated with molecular and phenotypic data, including copy number alterations, specific point mutations, gene-level mutations, protein expression, microRNA expression, and drug response data (IC50 values).

Since the number of cell lines in the CCLE is about one order of magnitude smaller than the number of TCGA samples, the power to detect significant associations is also much smaller. Nevertheless, we found 460 significant associations at a Bonferroni-corrected significance level of 0.05. For example, KEAP1 mutations in lung adenocarcinoma are associated with xenobiotic metabolism in both TCGA samples and CCLE cell lines (Figure 8), in agreement with the known role of the NRF2/KEAP1 pathway in the regulation of xenobiotic response [16]. Note that KEAP1 mutations appear to be associated with the activation of the pathway, as observed in [17].

## 3. Discussion

In this work, we systematically analyzed cancer metabolic subtypes, defined by clustering the patient samples using the transcription profiles of metabolic gene sets. These subtypes are highly specific for each individual metabolism used to define the gene sets, and show a myriad of significant associations with both molecular and phenotypic features of the tumors.

We present these associations as a resource for the community, Available online: https://metaminer.unito.it (accessed on 10 February 2022) which, we believe, will be useful in two related but distinct ways. One the one hand, our results highlight the key role of metabolic profiles in classifying individual patients in a way that is potentially useful to devise precision treatments. This is particularly compelling for tumors for which no established classification exists, but in which metabolic subtypes strongly associated with phenotypic characteristics can be found.

On the other hand, our results can help formulate hypotheses on the mechanisms underlying the associations between metabolic subtypes on one hand, and molecular and phenotypic features on the other. These can then be experimentally tested in cell lines or organoids. For example, while both miR-222 (see, e.g., ref. [18] for a recent review) and inositol phosphate metabolism [19,20] have been often associated with cancer progression, their association, which recurs in a large number of tumor types, to the best of our knowledge, has not been previously reported, and deserves a mechanistic investigation.

Some limitations of our study should be pointed out. The strongest is the correlative nature of our results, which would require experimental analyses to be interpreted in terms of causality; this limitation is shared by all retrospective analyses of primary tumors. Moreover, the results pertaining to some specific tumor types need to be interpreted with caution. In particular, tumor types obtained by the combination of more specific types (such as GBMLGG, LUNG, etc.) might generate many statistically significant results of limited biological interest: for example, since GBM and LGG strongly differ in survival probability, any gene set able to distinguish through clustering between these two types of tumors will be associated with survival. Moreover, some tumor types are characterized by mutations leading to widespread metabolic reprogramming (such as IDH1 mutations in glioma [21] and inactivation of the von Hippel–Lindau tumor suppressor in kidney cancer [22]). Therefore, in these tumors, many or most metabolic gene sets are associated with clinical or molecular features. Such associations, while statistically correct, might not lead to a manageable number of specific, testable mechanistic hypotheses. Finally, while k-medoid clustering has been shown to perform well in the classification of cancer samples based on bulk RNA-seq data [11], we did not explore how other clustering strategies would perform in detecting associations between metabolic subtypes and molecular/clinical fetaures.

## 4. Methods

### 4.1. Data

#### 4.1.1. TCGA

All TCGA data (release 28 February 2016) were obtained from the Broad TCGA GDAC site (https://gdac.broadinstitute.org/, accessed on 27 January 2021), by means of firehose_get, version: 0.4.1. The gene expression data clustered to generate the metabolic subtypes were the normalized gene-level transcript per million (TPM) values obtained with the RNA-Seq by the Expectation-Maximization (RSEM) method.

Copy number data were obtained from the segmented data using the cghMCR [23], DNAcopy [24], and CNTools [25] Bioconductor packages. The cghMCR package allows the calculation of segment gain or loss (SGOL) from segmented data, by means of a modified version of the Genomic Identification of Significant Targets in Cancer (GISTIC) algorithm. The *segment* function of the DNAcopy package was used to segment the normalized data so that chromosome regions with the same copy number had the same segment mean values. Then, using the *getRS* function from CNTools, the data returned by *segment* were organized in a matrix format. Finally, the *SGOL* function of cghMCR was used to compute gene-level SGOL scores by calculating the sum (parameter method) of all the positive and negative values, respectively, above and below a set threshold (0.5, −0.5). Finally, SGOL scores for all genes included in each Broad Institute positional gene set were averaged to obtain the SGOL score of each region.

Genome-wide DNA methylation was obtained from the gene-level summarized Human Methylome 450 k data, as provided by GDAC, then summed over all genes to associate a global DNA methylation value to each sample. miRNA expression was obtained from the Illumina HiSeq data, as deposited in GDAC. Specific point mutations for each sample were obtained from the “mutation packager oncotated calls” provided by GDAC, considering only non-synonymous mutations. For gene-level mutations, a sample was considered as mutated in a gene if it carried one or more specific point mutations associated with the gene.

Protein expression data produced by Reverse-Phase Protein microArrays (RPPA) were obtained from the corresponding files annotated with gene names from the GDAC site. Survival and other clinical data were obtained from the “Tier 1 clinical pick” files Available online: GDAC.

#### 4.1.2. CCLE

Gene expression data of 723 cell lines associated with one of the TCGA tumor types were obtained, as gene-level TPMs computed with RSEM, from the CCLE web site. We also retrieved from the CCLE web site the following data to be correlated with metabolic subtypes:Gene-level copy number data: the absolute score provided by CCLE was processed with the same steps used for the SGOL score reported for the TCGA data to obtain region-based copy number data;miRNA expression data (which were log-transformed for display in the figures);Somatic mutation data (here we considered only non-synonymous mutations);IC50 values for the available drugs;Metabolite levels.

#### 4.1.3. Metabolic Gene Sets

Gene sets were obtained from the c2.KEGG, c2.REACTOME, c5.BP, and Hallmark MSigDB v5.2 collections. Metabolic gene sets were defined as those whose name matched the string “metabol” but not the string “regul” (to include only gene sets directly involved in metabolic pathways rather than in their regulation). In this way, we obtained 345 metabolic gene sets (41, 35, 265, and 4 for the c2.KEGG, c2.REACTOME, c5.BP, and Hallmark collections, respectively).

### 4.2. Generation of Metabolic Clusters

For each tumor type and each metabolic gene set, a Duda–Hart test [12] was performed to detect cluster separation. This was conducted on both log-transformed and rank-transformed expression values. Since the latter method turned out to be more conservative, it was used in the following. For the gene set/tumor pairs for which the Duda–Hart test revealed the existence of a cluster structure (*p* < 0.001), the patients were divided into metabolic subtypes using partition around medoid (PAM) clustering [9] on the Spearman rank correlation coefficient-based distance matrix obtained from the expression of the genes in the gene set. The number of clusters (the maximum being set at 10) generated by each metabolic gene set was based on optimizing the average silhouette width. The normalized mutual information between two clusterings of the same patients based on different gene sets was computed with the R package *infotheo* [26], using the entropy of the empirical probability distribution and normalizing to the geometric mean of the entropies. Genes differentially expressed among clusters were identified using a Kruskal–Wallis test. The top 20 genes by *p*-value are shown in the heatmaps.

### 4.3. Statistical Tests of Association

The statistical tests used for establishing the association between each clinical/molecular feature and metabolic subtypes depend on the type of variable describing the feature. The metabolic subtype is always a categorical variable indicating the cluster membership of each patient.

For continuous variables (microRNA expression, DNA methylation, CNA SGOL scores) a we used Mann–Whitney U test (Kruskal–Wallis for k>2). In the case of microRNA expression, given the large number of microRNAs with expression equal to zero in each sample, the expression data were rank-transformed with random resolution of ties before the statistical test.For categorical variables (e.g., presence of a given mutation) we used Fisher’s exact test ($\chi^{2}$ test for tables larger than 2×2, removing levels for which the expected count was less than 5).For survival (RFS or OS), we used the log-rank test. The test was not considered valid when the expected number of events in any metabolic subtype was less than 5.

#### Multiple Testing

Bonferroni correction was applied separately to each variable type and to all tumor types together. Thus, for example, the Bonferroni correction for the association with miRNA expression takes into account all tests of association between all miRNAs and all metabolic gene sets across all tumor types. We chose to perform the correction separately for the feature types because these contain very different numbers of variables (hundreds of variables for miRNA expression and a single one for overall survival). Therefore, an overall Bonferroni correction would unduly penalize the variable types containing few variables.

## 5. Conclusions

The systematic analysis of the associations between transcriptomic-based metabolic subtypes and other clinical and molecular tumor features can serve as a guide to the formulation of mechanistic hypotheses for experimental validation, with the potential to improve our understanding of the role of metabolic alterations in cancer progression and to exploit metabolism-based stratification to develop new precision medicine strategies.

## Figures and Tables

**Figure 1 cancers-14-02145-f001:**
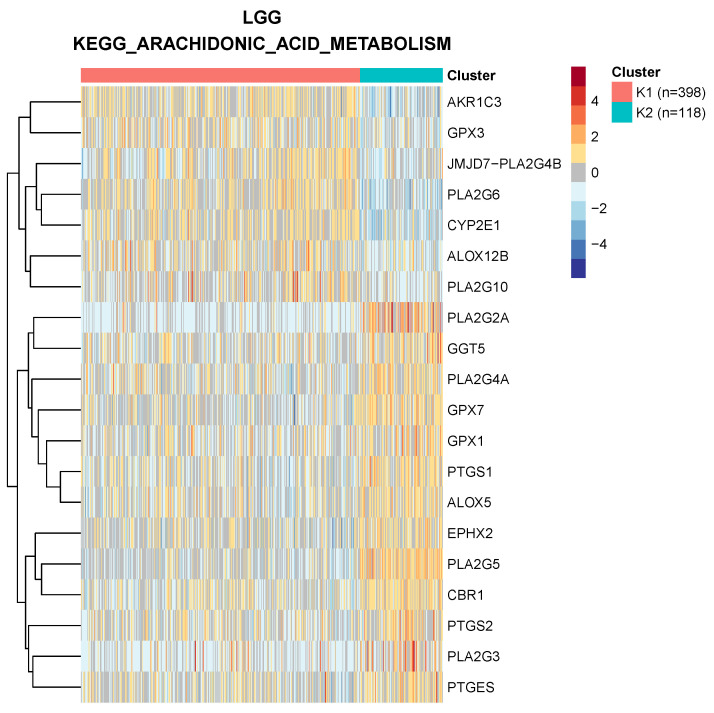
Clustering of low-grade glioma patients using the genes involved in arachidonic metabolism according to KEGG. Only the 20 genes most differentially expressed between the two clusters are shown.

**Figure 2 cancers-14-02145-f002:**
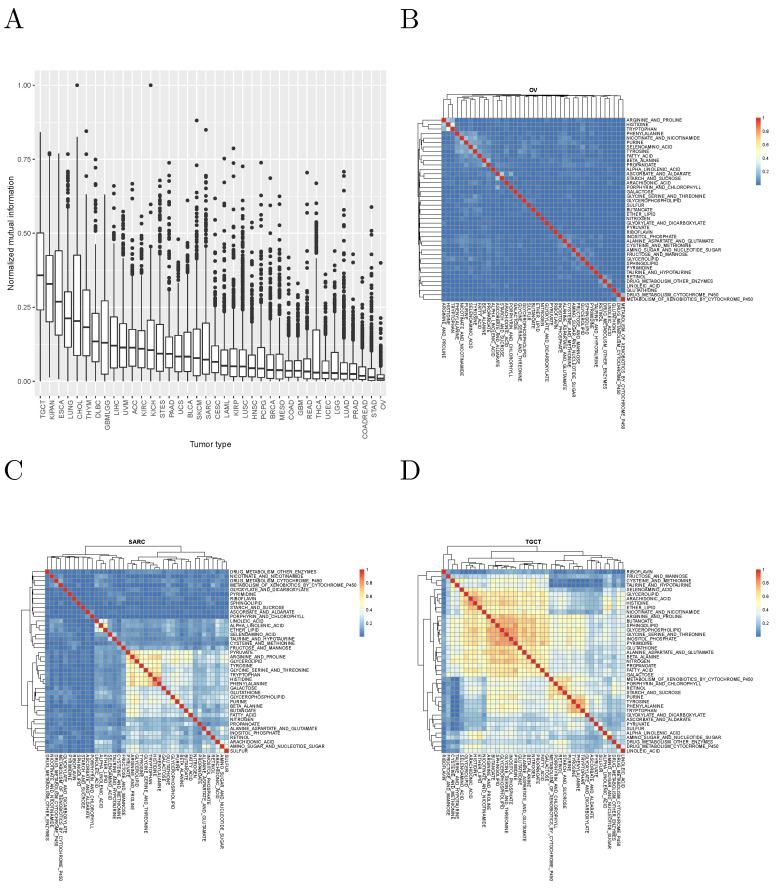
(**A**): Normalized mutual information between clusterings of samples using KEGG metabolic gene sets (tumor type abbreviations are those used by the TCGA and are explained in Appendix A). (**B**–**D**): Examples of tumor types with low, intermediate, and high normalized mutual information between clusterings obtained with different metabolic pathways.

**Figure 3 cancers-14-02145-f003:**
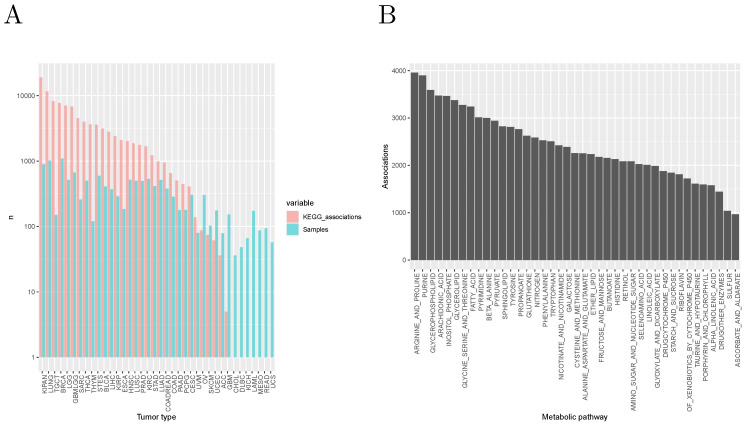
(**A**) Number of samples and of significant associations with molecular and clinical features by tumor type. (**B**) Number of significant associations by KEGG metabolic pathway.

**Figure 4 cancers-14-02145-f004:**
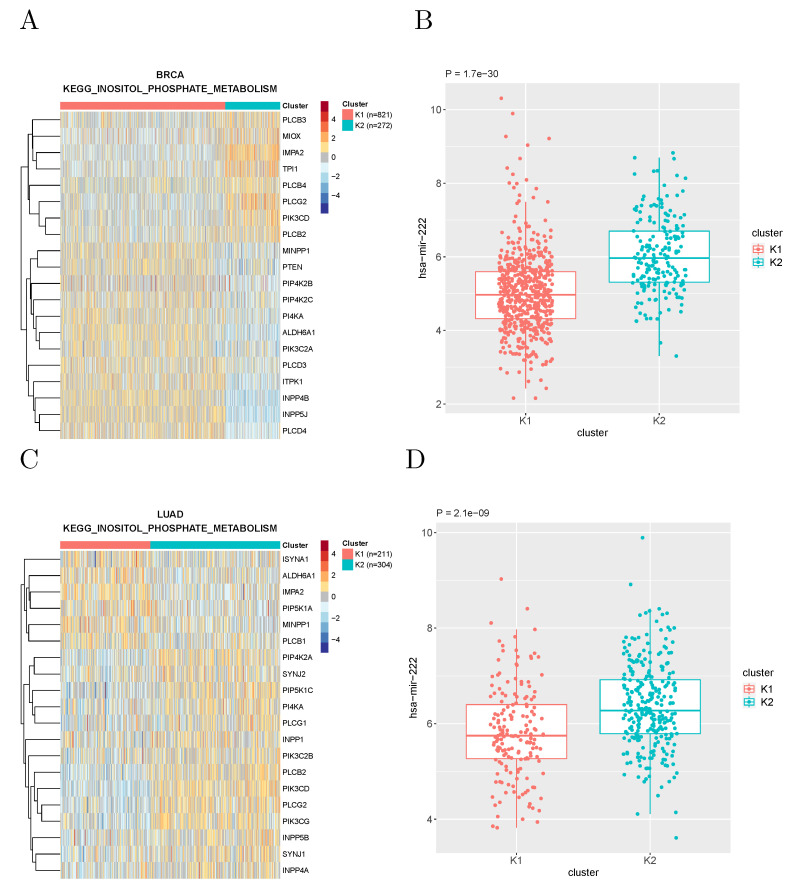
(**A**) Clustering of breast cancer samples using the genes associated to inositol phosphate metabolism according to KEGG. (**B**) Expression of miR-222 in the two clusters. (**C**,**D**) Same for lung adenocarcinoma samples.

**Figure 5 cancers-14-02145-f005:**
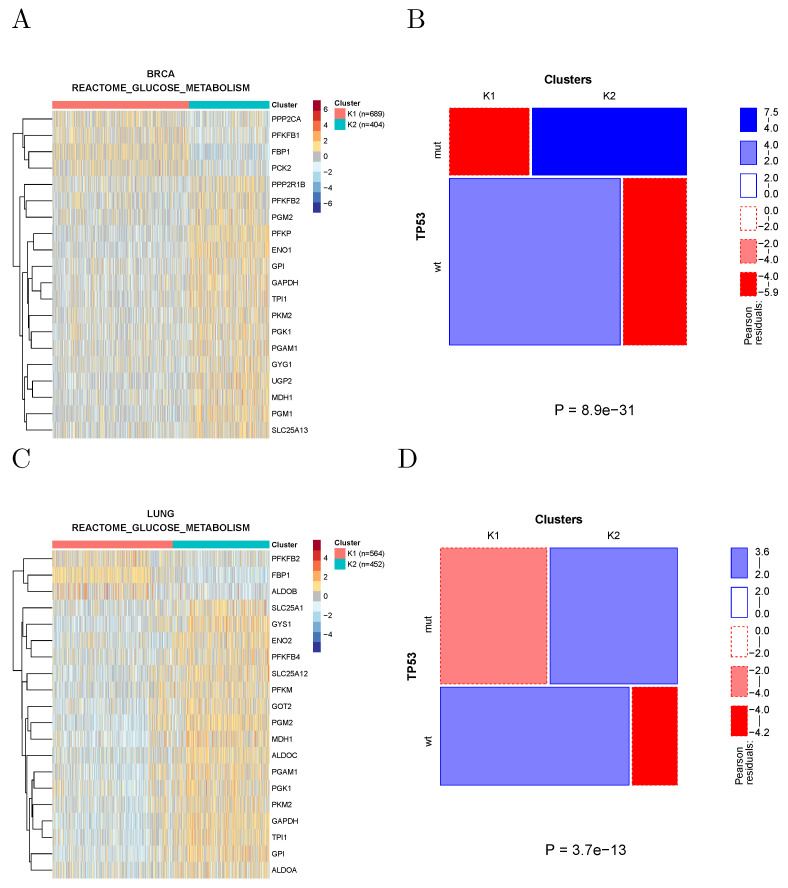
(**A**) Clustering of breast cancer samples using the genes associated with glucose metabolism according to REACTOME. Cluster K2 shows upregulation of Warburg effect genes, such as GAPDH, PGK1, and PKM2. (**B**) The prevalence of TP53-mutated samples is significantly higher in cluster K2. (**C**,**D**) Same for lung cancer samples.

**Figure 6 cancers-14-02145-f006:**
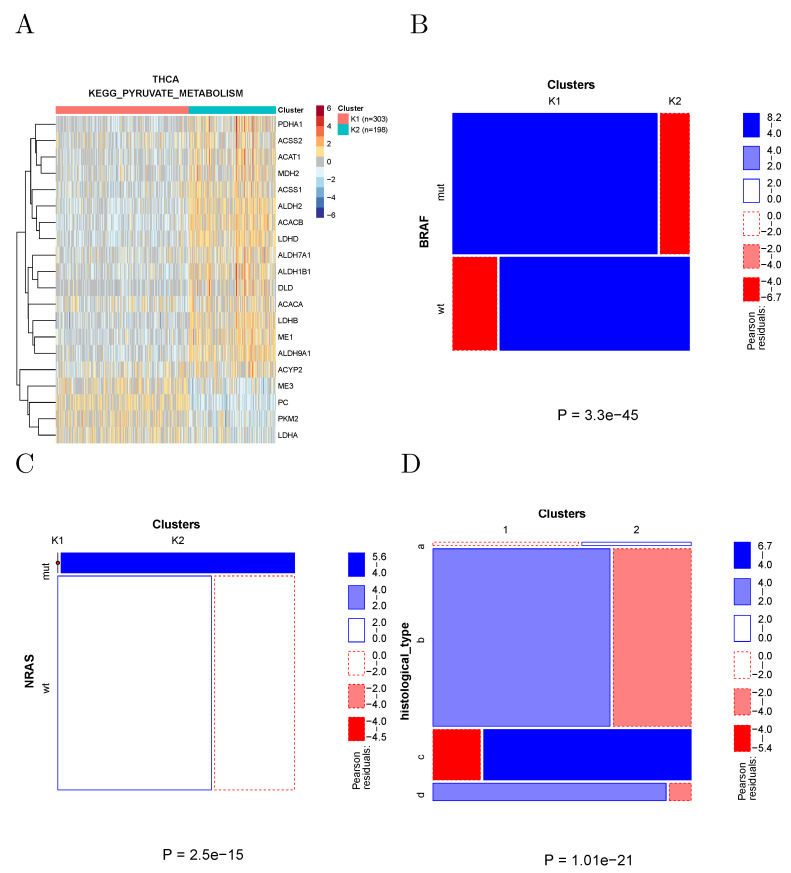
(**A**) Clustering of thyroid cancer samples using the genes associated with KEGG to pyruvate metabolism. (**B**) Most BRAF-mutated samples fell into Cluster 1. (**C**) All NRAS-mutated samples fell into Cluster 2. (**D**) The two clusters significantly differed in the prevalence of histological types; (a): other type; (b): thyroid papillary carcinoma—classical/usual; (c): thyroid papillary carcinoma—follicular (>= 99% follicular patterned); (d): thyroid papillary carcinoma—tall cell (>= 50% tall cell features).

**Figure 7 cancers-14-02145-f007:**
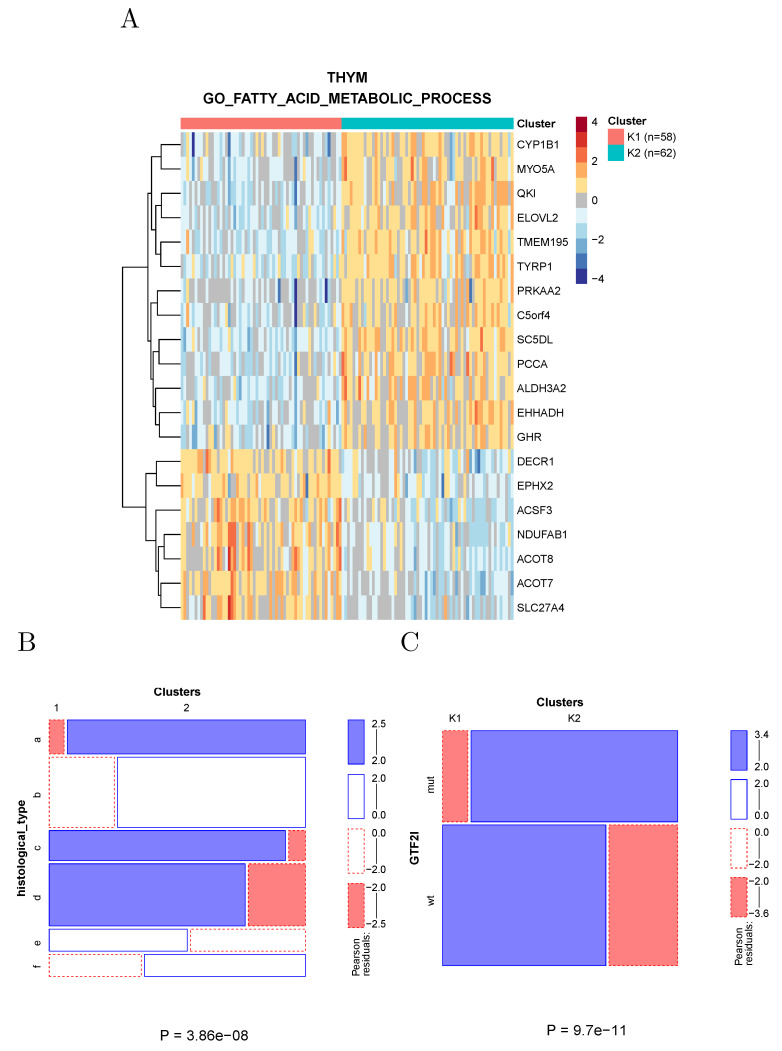
(**A**) Clustering of thymoma samples using the genes associated by KEGG to fatty acid metabolism. (**B**) The clusters significantly overlap the known histological types; (a): thymoma, type a; (b): thymoma, type ab; (c): thymoma, type b1; (d): thymoma, type b2; (e): thymoma, type b3; (f): thymoma, type c. (**C**) GTF2I mutations significantly segregate between the two clusters.

**Figure 8 cancers-14-02145-f008:**
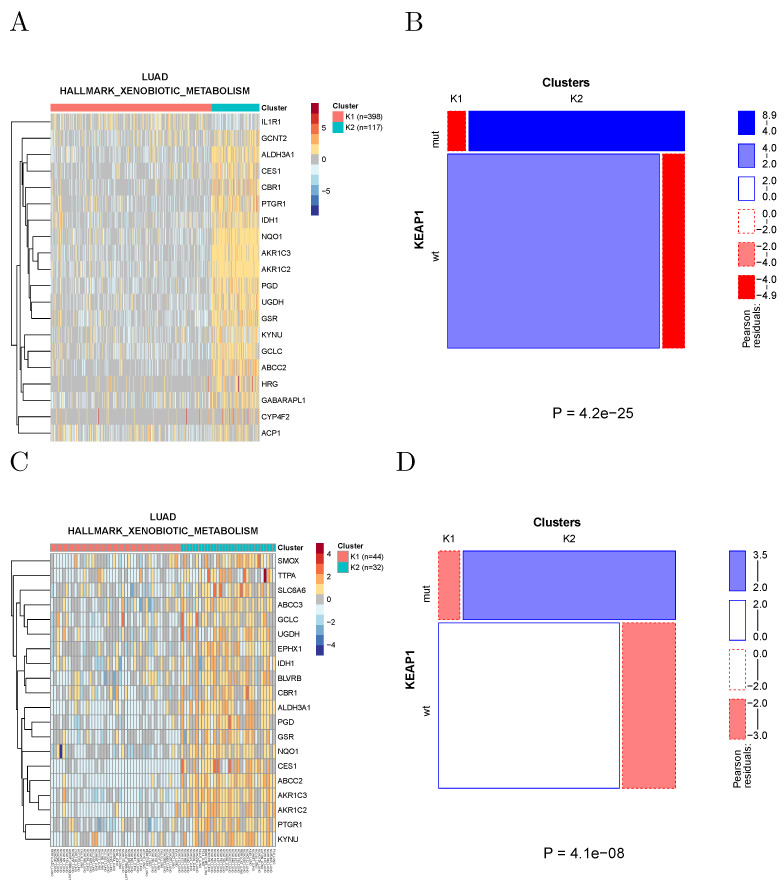
(**A**) Clustering of lung adenocarcinoma samples using the genes in the xenobiotic metabolism hallmark gene set. (**B**) Mutations of KEAP1 are strongly enriched in the K2 cluster where the pathway appears to be activated. (**C**) Clustering of lung adenocarcinoma cell lines using the same gene set. (**D**) Also in cell lines mutations are enriched in the cluster showing pathway activation.

## Data Availability

The results of the analysis are included as Appendix A and can be browsed interactively at https://metaminer.unito.it/, accessed on 10 February 2022.

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
