# Peer review of "Cancer Metabolic Subtypes and Their Association with Molecular and Clinical Features"

_cancers, 2022, doi:10.3390/cancers14092145_

Round 1
Reviewer 1 Report
Although there has been quite a few recent publications on multi-omic correlations of feature, this manuscript included clinical data transcriptomic data and epigenomic databases to identify metabolic pathways associated with metabolic phenotypes. Sufficient metabolomic datasets per se are missing from the TCGA tumor biopsy and CCLE cell line databases for a sufficient correlation. I believe this manuscript should check for grammar throughout and publish as is.
Author Response
Thank you for your favorable comments. We have thoroughly revised the manuscript for grammar and language.
Reviewer 2 Report
The authors introduce a framework for cancer phenotyping based on the clustering of genes associated with a metabolic gene set. They provide a database of the statistical associations between reported clusters and clinical features extracted from TCGA and CCLE data. Unfortunately, I do not consider the manuscript worthy to be published in Cancers.
I do not consider the methodological framework proposed as innovative. I have a few concerns as well as on the framework itself. The framework consists of identifying a cluster using Partition Around Medoids (PAM) method on each metabolic gene set. The number of clusters is defined by the silhouette algorithm. First, it seems that the choice of the number of clusters ranges between 2 and 10. The possibility of having no cluster separation has not been taken into account. Moreover, the authors did not evaluate different methods of clustering. Also, if PAM is “a more robust method compared to k-means”, it is only one of many clustering methods available and it is not even the best.
My concerns were not limited to the framework but also to the meaning of the results proposed. Looking the Supplementary Data 2, it seems that each clinical feature reported has a large number of gene sets statistically associated. For example, the clinical feature “Overall_survival” in the tumor type GBMLGG has been associated with 299 of 345. In this context, it is almost always possible to find an association with the overall survival in GBMLGG cancer. This can obviously lead to a misinterpretation of the results.
“Bonferroni correction was applied separately to each variable”. The threshold used to have a statistically associated correlation is not reported. From Supplementary Data 2, it seems the authors consider a corrected p-value of 0.05 as a threshold. This does not take into account that more than a million associations were tested. The exact number of associations is not reported.
Unfortunately, I was not able to evaluate the website https://metaminer.unito.it because it is not accessible.
Reviewer 3 Report
The manuscript entitled " Cancer metabolic subtypes and their association with molecular and clinical features" describes the use of a framework based on statistical associations, that will contribute to the understanding of the role of metabolic alterations in cancer and to the development of precision therapeutic strategies.
In my opinion the manuscript is well written and the methods explored and the authors fundamented the results. In this sense the manuscript should be accepted after minor revisions.
Comments:
- Abbreviation should be described whe used for the first time
- The font numeber of graphics shoul be increased to better visualization
- The caption of figures should be centered below the figure
Author Response
Thank you for your favorable comments.
We have explained all abbreviations used in the text, and enlarged the fonts in Figures 4-8. The legends have been aligned to the figures.
Round 2
Reviewer 2 Report
The authors replied adequately to my concerns. I do not have other comments.